# Earliest evidence for fruit consumption and potential seed dispersal by birds

Han Hu[1,2]*, Yan Wang[3]*, Paul G McDonald[2], Stephen Wroe[2], Jingmai K O'Connor[4,5,6], Alexander Bjarnason[1], Joseph J Bevitt[7], Xuwei Yin[8], Xiaoting Zheng[3,8], Zhonghe Zhou[5,6], Roger BJ Benson[1]

[1]Department of Earth Sciences, University of Oxford, Oxford, United Kingdom; [2]Zoology Division, School of Environmental and Rural Sciences, University of New England, Armidale, Australia; [3]Institute of Geology and Paleontology, Linyi University, Linyi, China; [4]Field Museum of Natural History, Chicago, United States; [5]Key Laboratory of Vertebrate Evolution and Human Origins, Institute of Vertebrate Paleontology and Paleoanthropology, Chinese Academy of Sciences, Beijing, China; [6]Chinese Academy of Sciences Center for Excellence in Life and Paleoenvironment, Beijing, China; [7]Australian Centre for Neutron Scattering, Australian Nuclear Science and Technology Organisation, Sydney, Australia; [8]Shandong Tianyu Museum of Nature, Linyi, China

*For correspondence:
han.hu@earth.ox.ac.uk (HH);
wangyan6696@lyu.edu.cn (YW)

**Competing interest:** The authors declare that no competing interests exist.

**Abstract** The Early Cretaceous diversification of birds was a major event in the history of terrestrial ecosystems, occurring during the earliest phase of the Cretaceous Terrestrial Revolution, long before the origin of the bird crown-group. Frugivorous birds play an important role in seed dispersal today. However, evidence of fruit consumption in early birds from outside the crown-group has been lacking. *Jeholornis* is one of the earliest-diverging birds, only slightly more crownward than *Archaeopteryx*, but its cranial anatomy has been poorly understood, limiting trophic information which may be gleaned from the skull. Originally hypothesised to be granivorous based on seeds preserved as gut contents, this interpretation has become controversial. We conducted high-resolution synchrotron tomography on an exquisitely preserved new skull of *Jeholornis*, revealing remarkable cranial plesiomorphies combined with a specialised rostrum. We use this to provide a near-complete cranial reconstruction of *Jeholornis*, and exclude the possibility that *Jeholornis* was granivorous, based on morphometric analyses of the mandible (3D) and cranium (2D), and comparisons with the 3D alimentary contents of extant birds. We show that *Jeholornis* provides the earliest evidence for fruit consumption in birds, and indicates that birds may have been recruited for seed dispersal during the earliest stages of the avian radiation. As mobile seed dispersers, early frugivorous birds could have expanded the scope for biotic dispersal in plants, and might therefore explain, at least in part, the subsequent evolutionary expansion of fruits, indicating a potential role of bird–plant interactions in the Cretaceous Terrestrial Revolution.

## Editor's evaluation

This article provides important new information on the ecology and morphology of a phylogenetically and temporally interesting early avialan. The work has important implications that should stimulate future research on Mesozoic bird-plant interactions.

## Introduction

Birds are among the most speciose extant vertebrate groups, playing unique ecological roles through their diverse flight and dietary adaptations (*Prum et al., 2015*). Crown-group birds include both specialised and opportunistic frugivores, that collectively are major consumers of fruits and important

**eLife digest** Birds and plants have a close relationship that has developed over millions of years. Birds became diverse and abundant around 135 million years ago. Shortly after, plants started developing new and different kinds of fruits. Today, fruit-eating birds help plants to reproduce by spreading seeds in their droppings. This suggests that birds and plants have coevolved, changing together over time. But it is not clear exactly how their relationship started.

One species that might hold the answers is an early bird species known as *Jeholornis*. It lived in China in the Early Cretaceous, around 120 million years ago. Palaeontologists have discovered preserved seeds inside its fossilised remains. The question is, how did they get there? Some birds eat seeds directly, cracking them open or grinding them up in the stomach to extract the nutrients inside. Other birds swallow seeds when they are eating fruit. If *Jeholornis* belonged to this second group, it could represent one of the early steps in plant-bird coevolution.

Hu et al. scanned and reconstructed a preserved *Jeholornis* skull and compared it to the skulls, especially the mandibles, of modern birds, including species that grind seeds, species that crack seeds and species that eat fruits, leaving the seeds whole. The analyses ruled out seed cracking. But it could not distinguish between seed grinding and fruit eating. Hu et al. therefore compared the seed remains found inside *Jeholornis* fossils to seeds eaten by modern birds. The fossilised seeds were intact and showed no evidence of grinding. This suggests that *Jeholornis* ate whole fruits for at least part of the year.

At around the time *Jeholornis* was alive, the world was entering a phase called the Cretaceous Terrestrial Revolution, which was characterized by an explosion of new species and an expansion of both flowering plants and birds. This finding opens new avenues for scientists to explore how plant and birds might have evolved together. Similar analyses could unlock new information about how other species interacted with their environments.

agents of seed dispersal. However, the occurrence of fruit consumption among early birds, outside the crown-group, is not yet clear. The early ecological diversification of birds in the Early Cretaceous (>130 Ma) (*Yang et al., 2020*) was a landmark event in the evolution of terrestrial ecosystems, adding considerably to species richness of terrestrial ecosystems (*Benson, 2018a*; *Yu et al., 2021*), and with impacts on the evolutionary histories of other flying groups (*Benson et al., 2014b*; *Clapham and Karr, 2012*). This was followed by a considerable long-term expansion of the abundance and disparity of fruits and fruit-like structures through much of the Cretaceous (*Eriksson et al., 2000a*; *Eriksson, 2008*), as part of the major floral transition from gymnosperm- to angiosperm-dominated floras that is often referred to as the 'Cretaceous Terrestrial Revolution' (KTR) (*Benton, 2010*; *Lloyd et al., 2008*). A macroevolutionary connection between early birds and this important event of fruit evolution has been suggested (*Pejchar et al., 2008*; *Sekercioglu, 2006*; *Tiffney, 2004*), but is so far unsubstantiated by fossil evidence of fruit consumption by early birds, limiting our understanding of the evolutionary origins of an important aspect of plant–animal interactions.

The Jeholornithiformes from the Early Cretaceous Jehol Biota of China are one of the earliest-diverging avian lineages and are morphologically very distinct from crown-group birds, retaining an elongate, bony tail, which is absent in all other birds except for the Late Jurassic *Archaeopteryx* (*Wang et al., 2018*; *Zhou and Zhang, 2002*). They also possess several advanced, flight-related morphologies, suggesting a unique form of powered flight (*O'Connor et al., 2013*; *Zheng et al., 2020*; *Zhou and Zhang, 2002*). The most abundant jeholornithiform, *Jeholornis*, has been interpreted as the earliest granivorous bird, based on the reportedly 'deep' mandible and traces identified as seeds preserved in the abdominal area (*Zhou and Zhang, 2002*). Reduced dentition and the presence of a gastric mill further suggest a herbivorous diet (*O'Connor et al., 2018*). However, there is no consensus on whether seeds entered the gut of *Jeholornis*, and other early birds, through deliberate and destructive seed consumption (granivory), or through consumption of fleshy propagules such as true angiosperm fruits or gymnosperm arils (herein referred to as 'fruit consumption' for convenience, encompassing both consumption of all types of fleshy diaspores, not limited to true fruits) (*Ksepka et al., 2019*; *Mayr et al., 2020*; *O'Connor, 2019*; *O'Connor et al., 2018*; *O'Connor and Zhou, 2020*). Indeed, a recent review identified these as 'seed meals' without clarification (*Miller and*

*Pittman, 2021*). Clarifying between these two hypotheses has significant implications with regard to the early evolution of bird–plant interactions, because fruit consumption could result in beneficial co-evolutionary mutualism, whereas seed consumption does not. This therefore is relevant to understanding whether early birds could have been important agents of seed dispersal with a potential mutualistic co-evolutionary influence on plant evolution during the KTR.

Interpretations regarding diet in *Jeholornis* and other potentially granivorous early birds (*Ksepka et al., 2019*; *Zheng et al., 2018*; *Zheng et al., 2011*) have previously been framed using qualitative observations and subjective assessments, with minimal formal comparison to extant species, and in the absence of a detailed understanding of jeholornithiform cranial anatomy (*Lefèvre et al., 2014*; *O'Connor et al., 2012*; *O'Connor et al., 2018*; *Zhou and Zhang, 2003*; *Zhou and Zhang, 2002*). We here report an exquisitely preserved new *Jeholornis* specimen, STM 3–8, from the Shandong Tianyu Museum of Nature, Pingyi, China. We use high quality three-dimensional (3D) data acquired through the synchrotron tomography to reveal the key cranial features of this taxon and build a precise and almost complete cranial reconstruction of this key stem bird. This information is used to test and determine the two diet hypotheses of *Jeholornis*, through geometric morphometric (GMM) analyses of the mandible (3D) and cranium (2D), and high-resolution computed tomography (CT) 3D visualisations of the alimentary contents of extant birds. Our approach demonstrates the importance of applying multiple methods simultaneously to solve complex palaeoecological questions.

## Results

### Cranial anatomy

*Jeholornis* has been frequently studied and cited because of its key phylogenetic position, and many specimens are known. However, because specimens are often compressed, and are preserved in slabs, little unequivocal cranial information has been available (*Lefèvre et al., 2014*; *O'Connor et al., 2012*; *O'Connor et al., 2018*; *O'Connor et al., 2013*; *Wang et al., 2020*; *Zheng et al., 2020*; *Zhou and Zhang, 2003*; *Zhou and Zhang, 2002*). Our 3D reconstruction of the exquisitely preserved skull of *Jeholornis* STM 3–8 (*Figure 1*; *Figure 1—figure supplement 1*; for detailed taxonomic information see Supplementary Information) reveals that *Jeholornis* retains a plesiomorphic diapsid skull, and provides considerable new anatomical data.

Although an unfused postorbital was previously inferred based on the basal phylogenetic position of *Jeholornis* (*Wang and Hu, 2017*), STM 3–8 provides the first direct evidence of this. The postorbital is proportionally large with a well-developed jugal process that contacts the jugal, forming a robust, complete postorbital bar (*Figure 1*). This is a plesiomorphy shared with non-avian theropods and other stem birds including *Archaeopteryx* and *Sapeornis* (*Hu et al., 2020a*; *Rauhut et al., 2018*), contrasting with the reduced or absent postorbital bar in the Ornithothoraces including modern birds (*Hu et al., 2020b*). The squamosal possesses a postorbital process that likely contacted the postorbital to form the supratemporal arch. The ventral process of the squamosal is short and would not have contacted the quadratojugal. The squamosal of *Jeholornis* is remarkably anteroposteriorly broad even compared to that of *Archaeopteryx* (*Rauhut, 2014*; *Rauhut et al., 2018*). A complete bony upper temporal bar is supposed to exist based on the articular facet in the postorbital, while this bar is broken and probably linked by ligament in Late Cretaceous bird *Ichthyornis* (*Field et al., 2018*).

The palatal complex is nearly completely preserved, including the palatine, pterygoid, and vomer; the absence of the ectopterygoid is most likely preservational (*Figure 1*). The palate of *Jeholornis* exhibits few modifications from the non-avian theropod condition, and closely resembles that of *Archaeopteryx* (*Elzanowski and Wellnhofer, 1996*; *Mayr et al., 2007*; *Rauhut et al., 2018*). The palatine is broad with a well-developed jugal process that contacts the maxilla. The pterygoid is elongated with no sign of the shortening that occurs in more derived birds and the pterygoid flange is well developed, indicating the presence of an ectopterygoid. The vomer is dorsoventrally thin with bifurcated caudal flanges oriented nearly vertical to the rostral body, similar to the condition in *Sapeornis* (*Hu et al., 2019*).

While the temporal and palatal regions retain plesiomorphies, the rostrum of *Jeholornis* is heavily modified. The new specimen reveals that its premaxillae corpora are fused while the frontal processes remain separate. Rostral fusion of the premaxillae is also present in extant birds, confuciusornithiforms and several enantiornithines for example *Linyiornis* and *Shangyang* (*Wang and Zhou, 2019*; *Wang*

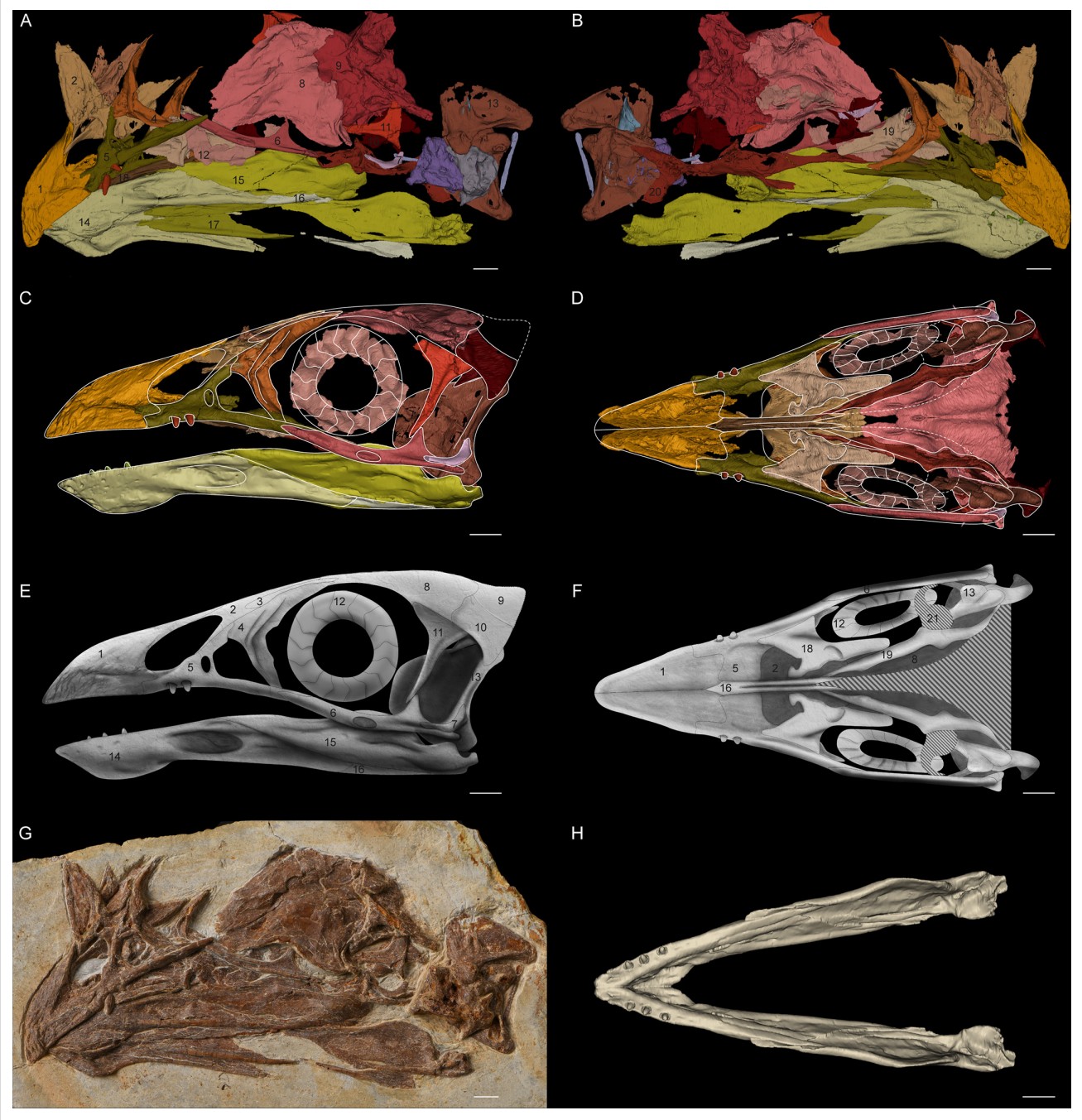

**Figure 1.** *Jeholornis* STM 3–8. (**A**) Left and (**B**) right views of the 3D reconstructed model of the skull. (**C**) Left and (**D**) ventral views of the reassembled 3D model of the skull. (**E**) Left and (**F**) ventral views of the 2D cranial reconstruction. (**G**) Photograph of the skull. (**H**) Dorsal view of the reassembled 3D model of the mandible. Abbreviations: 1. premaxilla; 2. nasal; 3. preorbital ossification; 4. lacrimal; 5. maxilla; 6. jugal; 7. quadratojugal; 8. frontal; 9. braincase; 10. squamosal; 11. postorbital; 12. scleral ring; 13. quadrate; 14. dentary; 15. surangular; 16. angular; 17. splenial; 18. vomer; 19. palatine; 20. pterygoid; 21. potential ectopterygoid. Different bones are indicated by different colours. Dashed lines indicate the elements not preserved but suspected to exist. Scale bar equals 5 mm.

The online version of this article includes the following figure supplement(s) for figure 1:

**Figure supplement 1.** Photograph of the whole slab of *Jeholornis* STM 3–8.

*et al., 2016*). Its occurrence in *Jeholornis* indicates that rostral fusion of premaxillae evolved phylogenetically deeper among birds than previously thought. *Jeholornis* also shows dental reduction, with an edentulous premaxilla, two rostrally restricted maxillary teeth and three extremely tiny teeth in the dentary (*O'Connor and Zhou, 2020*; *Zhou and Zhang, 2002*; *Figure 1*).

## GMM analyses

We digitally reassembled the cranium and mandible of *Jeholornis* STM 3–8, producing 2D cranial and 3D mandible reconstructions (*Figure 1*). These were included in a 3D GMM analysis of the mandible and a 2D analysis of the cranium of extant birds and select extinct pennaraptorans (for landmark definitions see *Figure 2—figure supplement 1* and *Figure 2—source data 1*, *Figure 2—source data 2*), to evaluate the similarity of the mandible and cranium of *Jeholornis* to extant birds with different diets. Our main analysis is intended to test how seeds entered the gut of *Jeholornis* by distinguishing between two hypotheses, either (1) fruit consumption or (2) seed consumption (*Figure 2*, *Figure 2—figure supplement 2*). For this analysis, diets of extant birds were separated into five categories: (1) Seed-crackers (parrots): granivores that de-husk and fragment seeds using the beak prior to ingestion; (2) Seed-crackers (passerines): granivores that de-husk but do not extensively fragment seeds using the beak prior to ingestion; (3) Seed-grinders: granivores that primarily process seeds using a gastric mill, with minimal beak processing; (4) Fruit eaters; and (5) Other diets (such as folivores, carnivores, and omnivores). Our supplemental analysis includes a further split of 'Other diets', separating the 'Other diets' category into: (1) Probing for invertebrates; (2) Grabbing/pecking for invertebrates (*Figure 2—figure supplement 3*); (3) Piscivores; (4) Animal-dominated omnivores; (5) Carnivores (*Figure 2—figure supplement 4*); (6) Nectarivores; (7) Omnivores; (8) Plant-dominated omnivores (*Figure 2—figure supplement 5*). Our expectation is that these analyses will not provide an unambiguous classification of the diet of *Jeholornis* on their own, because craniomandibular shape data do not completely differentiate among diets in birds (*Navalón et al., 2019*), but that they may be capable of ruling out the occurrence of certain diets.

## Mandibular morphospace

The principal components analysis (PCA) results reveal that a large portion of mandibular shape variation (PC1: 38.16%) is related to the relative length of the mandible compared to its rostral depth: positive values of PC1 indicate short, deep mandibles, whereas negative values indicate long, low mandibles. PC2 explains 32.98% of variation and is also related to the relative depth of the mandible, with positive values indicating low mandibles with coronoid eminence absent or less developed, and negative values indicating deep mandibles with a large coronoid eminence. PC3 (10.25% of variation) is related to the curvature, with positive values indicating a straight profile in lateral view, and negative values indicating rostroventral curvature of the rostral portion of the mandible (*Figure 2A, B*).

The results plot *Jeholornis* near the centre of mandibular morphospace. Seed-crackers, especially parrots, are clearly separated from the other diet types including *Jeholornis* in mandibular morphospace (*Figure 2A, B*). They occupy a distinct region with high, positive values of PC1 and low, negative values of PC2, reflecting their deep and anteroposteriorly short mandibles with a large coronoid process and deep mandibular symphysis, which suits their seed-cracking diet by reducing the beak failure risk during cracking (*Soons et al., 2015*; *Soons et al., 2010*). The frugivorous parrot – *Psittrichas fulgidus* (*Billerman et al., 2020*) – has a shallow mandible compared to those seed-cracking parrots, and plots closer to the distribution of non-parrots, consistent with the hypothesis that species can secondarily lose specialisations associated with their ancestral diet.

Seed-cracking passerines also occupy an area with negative PC2 values compared to most frugivores and seed-grinders, being closer to seed-cracking parrots (*Figure 2A, B*). They also show negative values of PC3, indicating that they have more downward inclined mandibles, which is related to their ability to de-husk seeds (*van der Meij and Bout, 2008*). Therefore, finches are also clearly distinct from the position of *Jeholornis* in mandibular morphospace (*Figure 2A, B*), rejecting the previous hypothesis of *Jeholornis* as a seed-cracker (both parrot- and finch-type) (*Zhou and Zhang, 2002*).

*Jeholornis* is plotted within the overlapping range of frugivores, seed-grinders, and birds with 'other diets' in our main analysis (*Figure 2A, B*). Frugivores and seed-grinders show wide and highly overlapping distributions (*Figure 2A, B*), indicating that 'seed-grinding' granivores, which do not engage in pre-processing of seeds using the beak, exhibit little specialisation of mandibular

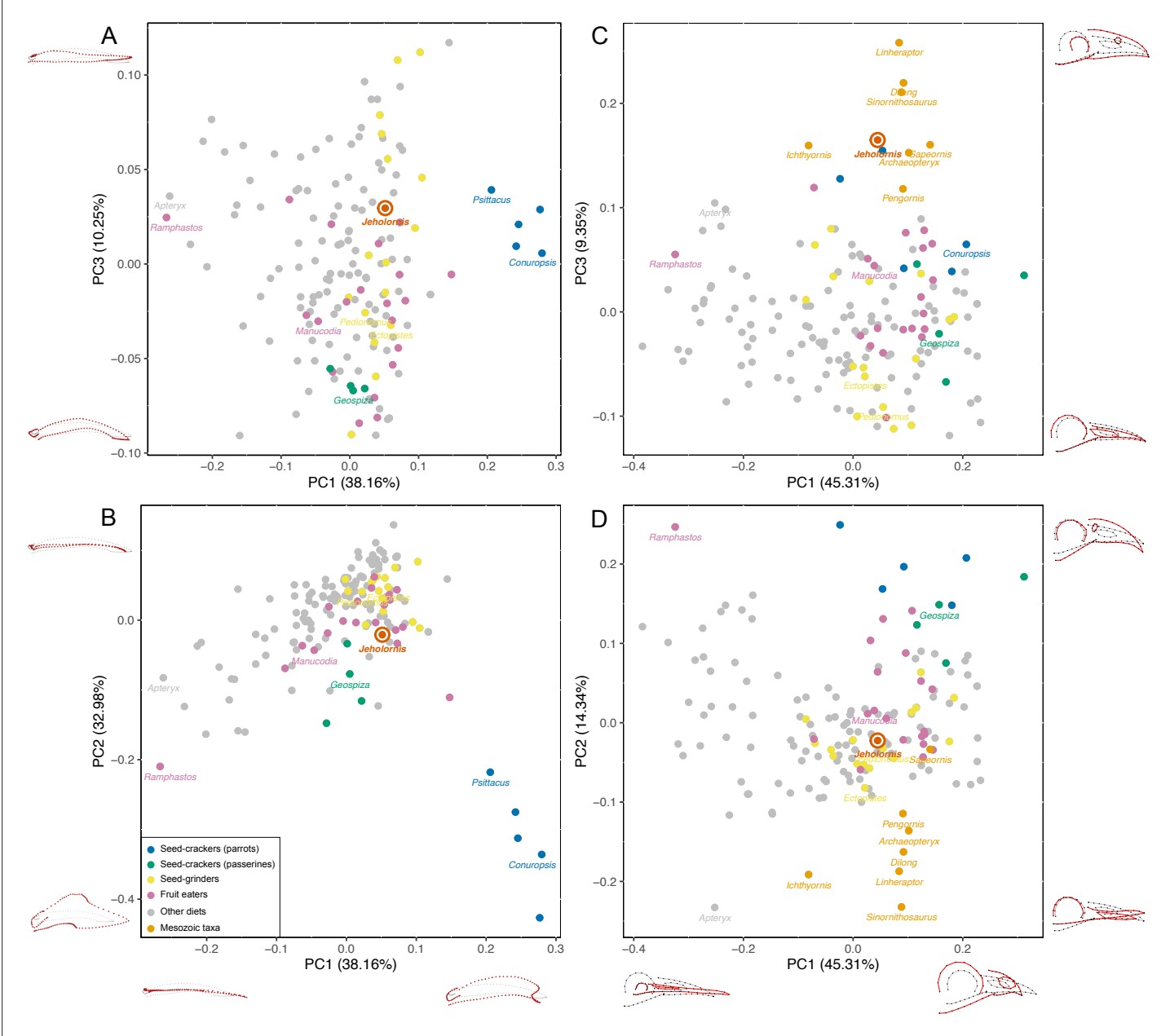

**Figure 2.** PCA result of 3D mandible shape (**A, B**) and 2D skull shape (**C, D**) with the diets of extant birds divided into Seed-crackers (parrots), Seed-crackers (passerines), Seed-grinders, Fruit eaters, and Other diets. Different diet categories are indicated by different colours, and key samples are labelled with generic names.

The online version of this article includes the following source data and figure supplement(s) for figure 2:

**Source data 1.** Descriptions of cranial and upper jaw landmarks and semi-landmarks (following *Bjarnason and Benson, 2021*).

**Source data 2.** Descriptions of mandible landmarks and semi-landmarks (following *Bjarnason and Benson, 2021*).

**Source data 3.** Euclidean distances in the full multivariate shape space of the mandible shape analysis.

**Source data 4.** Euclidean distances in the full multivariate shape space of the skull shape analysis.

**Figure supplement 1.** Landmark and semi-landmark locations in *Menura novaehollandiae* as an example of modern taxa used in geometric morphometric (GMM) analyses (following *Bjarnason and Benson, 2021*).

**Figure supplement 2.** PCA result of 3D mandible (**A, B**) and 2D skull shape (**C, D**) with the diets of extant birds divided into Seed-crackers (parrots), Seed-crackers (passerines), Seed-grinders, Fruit eaters, and Other diets and all the generic names labelled.

*Figure 2 continued on next page*

*Figure 2 continued*

**Figure supplement 3.** PCA result of 3D mandible (**A, B**) and 2D skull shape (**C, D**) with the diets of extant birds divided into Probing for invertebrates, Grabbing/pecking for invertebrates, and Other diets.

**Figure supplement 4.** PCA result of 3D mandible (**A, B**) and 2D skull shape (**C, D**) with the diets of extant birds divided into Piscivores, Animal-dominated omnivores, Carnivores, and Other diets.

**Figure supplement 5.** PCA result of 3D mandible (**A, B**) and 2D skull shape (**C, D**) with the diets of extant birds divided into Nectarivores, Omnivores, Plant-dominated omnivores, and Other diets.

---

morphology compared to 'seed-cracking' granivores. Therefore, although our results exclude *Jeholornis* from being a seed-cracker, they cannot distinguish between the hypotheses that seeds entered the gut of *Jeholornis* due to fruit consumption, or due to seed-grinding granivory.

Our supplemental analyses find that *Jeholornis* was unlikely to have had a probing or piscivorous diet; probing birds occupy negative PC1 values (*Figure 2—figure supplement 3*), and piscivores occupy positive PC2 values (*Figure 2—figure supplement 4*). However, *Jeholornis* cannot readily be distinguished from other diets such as the grabbing/pecking for invertebrates and omnivory (*Figure 2—figure supplements 3–5*). Euclidean distances in the full multivariate shape space suggest that the mandible of *Jeholornis* is relatively similar to those of various omnivorous (e.g. *Podica*), seed-grinding (e.g. *Calandrella*), frugivorous (e.g. *Crax*), and invertebrate pecking (e.g. *Picus*) birds (*Figure 2—source data 3*).

### Cranial morphospace

Cranial shape distinguishes between our focal diet categories less effectively than mandibular shape (*Figure 2C, D*, *Figure 2—figure supplement 2C, D*). Nevertheless, some separation is still evident, especially between seed-crackers and other dietary groups. This also indicates that *Jeholornis* was not a seed-cracking granivore. Extant seed-crackers occupy positive values of both PC1 and PC2, compared to more centrally positioned frugivores and seed-grinders. Variation in PC1 (45.31%) is related to the relative length of the rostrum compared to the jugal bar, with positive values indicating a shorter rostrum. Variation in PC2 (14.34%) is related to the depth and curvature of the rostrum, with positive values indicating deeper and rostroventrally curved rostra, present in seed-crackers and toucans (*Ramphastos*, which differs from seed-crackers in having a negative PC1 score). Variation in PC3 (9.35%) is related to the relative size of the orbit and naris, with positive values indicating smaller orbits and naris. Because some fossil samples included in our analyses are incomplete, we did not include the skull roof in this analysis. Our results indicate that seed-crackers have relatively short, deep and rostroventrally curved rostra compared to most other birds, including *Jeholornis*, *Sapeornis*, and other Mesozoic taxa.

Similar to the results of the mandible analyses, the results of the supplemental analyses of cranial shape also exclude *Jeholornis* from possessing a probing or piscivorous diet; probing birds occupy negative PC1 values (*Figure 2—figure supplement 3*), and piscivores occupy positive PC2 values (*Figure 2—figure supplement 4*).The other diets are also not readily distinguishable in the supplemental analyses of cranial shape (*Figure 2—figure supplements 3–5*). Euclidean distances in the multivariate shape space, excluding PC3 (which describes the large-scale differences between stem- and crown-group birds) suggest that the cranium of *Jeholornis* is similar to those of various frugivorous (e.g. *Manucodia*), seed-grinding (e.g. *Pedionomus*), and invertebrate pecking (e.g. *Hymenops*) birds (*Figure 2—source data 4*).

Mesozoic taxa are mostly separated from modern birds along PC2 and PC3, occupying negative values of PC2 and positive values of PC3 separately (*Figure 2C, D*). Among them, *Jeholornis* and *Sapeornis* are more similar to extant birds along PC2, which describes rostral morphology. This may reflect the dietary specialisation of *Jeholornis* and *Sapeornis* (as fruit or seed consumers) compared to other Mesozoic taxa. Nevertheless, they cluster with other Mesozoic taxa along cranial PC3, indicating conservative aspects shared with non-avian theropods, especially a proportionally small orbit and external naris.

### Alimentary content analyses

Our morphometric analyses indicate that *Jeholornis* was not a 'seed-cracker', but do not distinguish between frugivorous and seed-grinding granivorous diets. We therefore conducted a comparison of

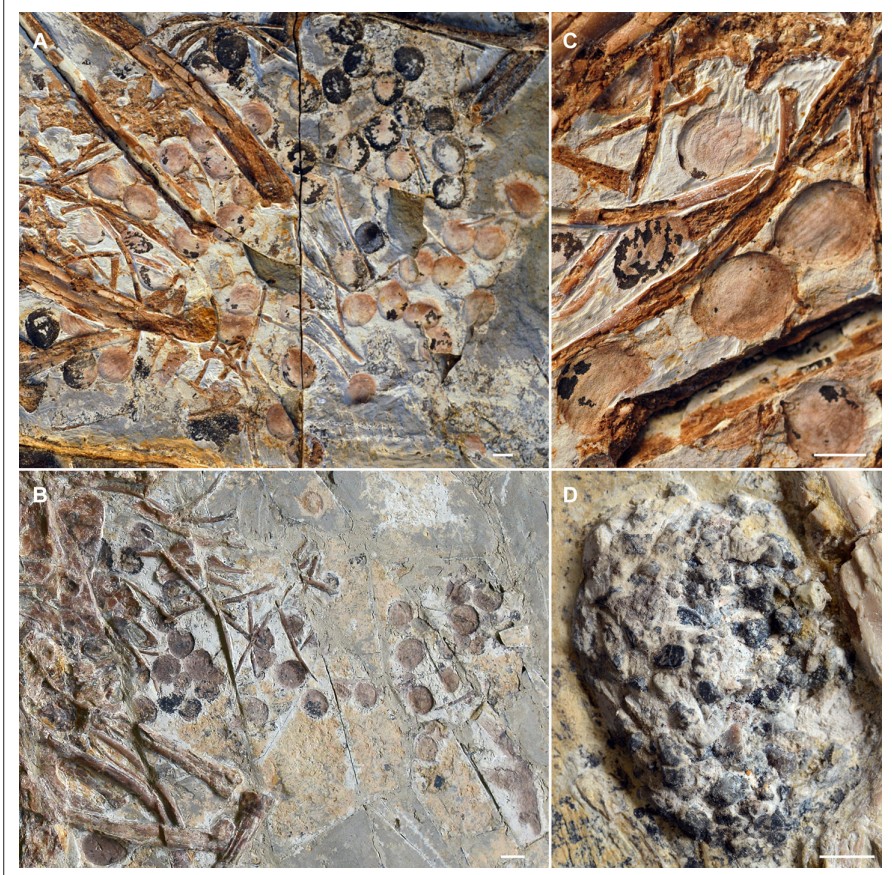

**Figure 3.** Seeds preserved in the abdominal area of selected *Jeholornis prima* specimens. (**A**) IVPP V13274 (holotype). (**B**) STM 2–41. (**C**) Close-up image of seeds in IVPP V13274 (**A**). (**D**) Gastrolith mass in *J. prima* STM 2–15. Photos in A–D followed figures in **O'Connor et al., 2018**. Scale bars equal 5 mm.

the alimentary contents of *Jeholornis* (**Figure 3**) with selected modern birds (**Figure 4**) using high-resolution CT scanning. Our modern bird sample includes frugivores (*Manucodia comrii*, Curl-crested manucode; *Bombycilla garrulus*, Bohemian waxwing), seed-cracking parrots (*Conuropsis carolinensis*, Carolina parakeet), seed-cracking passerines (*Geospiza fuliginosa*, Small ground-finch; *Calcarius lapponicus*, Lapland longspur), and seed-grinding granivores (*Ectopistes migratorius*, Passenger pigeon; *Pedionomus torquatus*, Plains-wanderer; *Thinocorus rumicivorus*, Least seedsnipe) (detailed specimen information see **Figure 4—source data 1**; detailed descriptions of their alimentary contents see Materials and methods).

Comparative evidence from those modern avian gut contents show that destructive seed consumption (seed predation) is strongly indicated by fragmentation (in seed-crackers) or abrasion (in seed-grinders) of seeds in the alimentary canals, which is likely a prerequisite for nutrient extraction. The seed remains are highly fragmented in seed-cracking parrots (**Figure 4E**), whereas in seed-cracking passerines, although the crop contents are almost intact, those in the stomach are also highly fragmentary (**Figure 4D**, **Figure 4—figure supplement 1E, F**). This is consistent with behavioural observations of finches and other granivorous passerines (**Billerman et al., 2020**), in which seed-cracking passerines use the beak only to remove the outer coats of seeds, and do not fragment the seed before ingestion, differing from parrots that can fragment seeds prior to ingestion (**Figure 4E**). Fragmentation of seeds in passerines is primarily achieved through the gastric mill, similar to some seed-grinders for example *E. migratorius* (Passenger pigeon) (**Figure 4C**, **Figure 4—figure supplement 1B**). However,

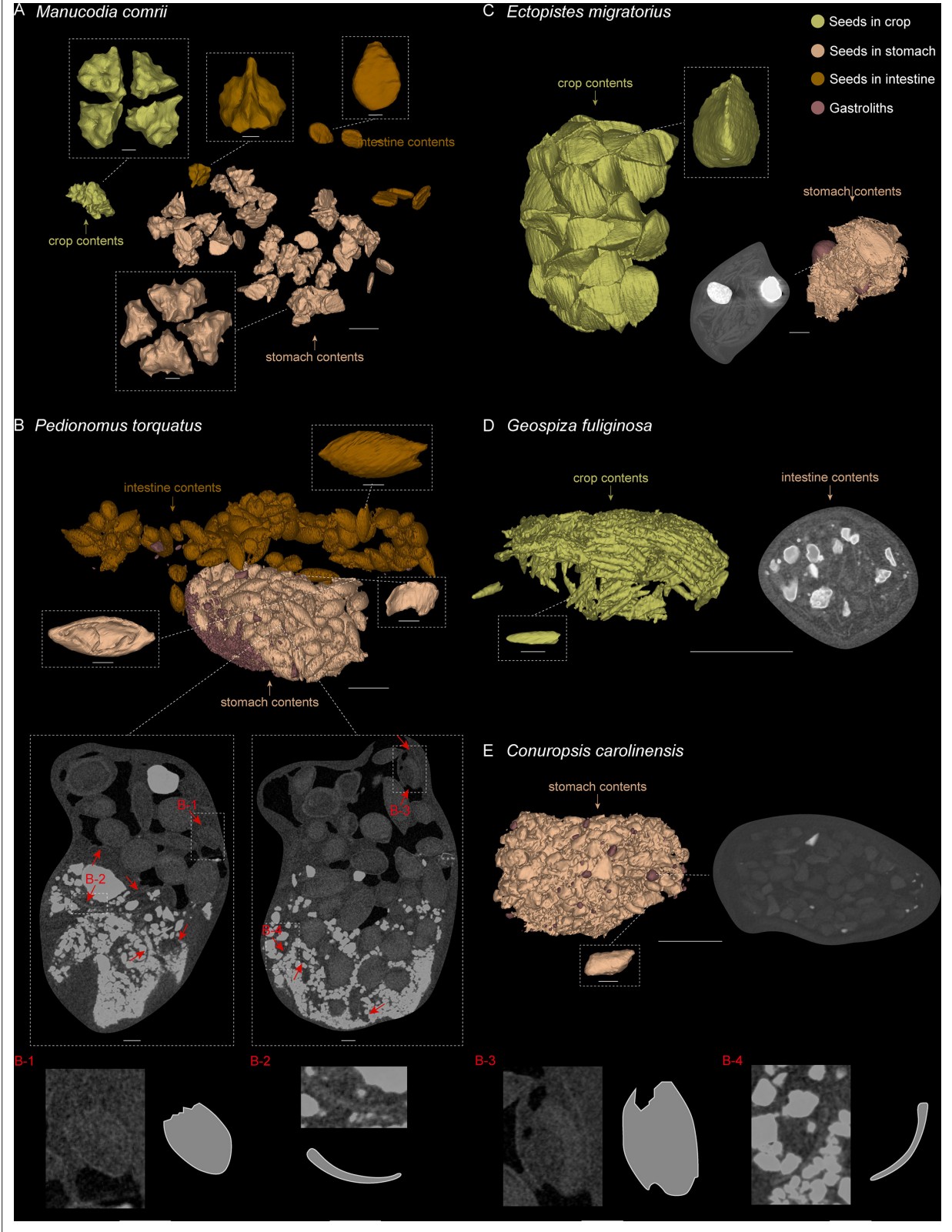

**Figure 4.** 3D reconstructed seed models preserved in alimentary tract of selected modern birds. (**A**) *Manucodia comrii* (fruit eater). (**B**) *Pedionomus torquatus* (seed-grinder). (**C**) *Ectopistes migratorius* (seed-grinder). (**D**) *Geospiza fuliginosa* (use both seed-cracking and seed-grinding strategies). (**E**) *Conuropsis carolinensis* (seed-cracker). Dash-lined boxes indicate local magnifications. Gastroliths are remarkably brighter than other contents in the

*Figure 4 continued on next page*

*Figure 4 continued*

slices. Red arrows indicate the breakages of seeds in slices, which are difficult to show in the reconstructed models. Scale bars equal 5 mm for the whole models and slices, and 1 mm for the magnification boxes.

The online version of this article includes the following source data and figure supplement(s) for figure 4:

**Source data 1.** Specimens used in the alimentary content analyses.

**Figure supplement 1.** Scanning slices of the alimentary contents in involved modern bird samples.

**Figure supplement 2.** Scanning slices of the alimentary contents in involved modern bird samples.

in most seed-grinders the gut contents consist of abraded and partially damaged, rather than highly fragmented, seed remains (*Figure 4B*, *Figure 4—figure supplement 2A–F*).

Seed remains in all the sampled granivores were tightly aggregated together, and typically co-occurred with gastroliths (*Figure 4B–E*). Gastroliths are especially abundant in some seed-grinders and seed-cracking passerines (*Figure 4B, D*) compared to the parrot (*Figure 4E*) and pigeon (*Figure 4C*). In contrast, the seed remains in frugivores are completely intact, often in their original 'within-fruit' configurations. They are sparsely dispersed in the alimentary tract, sometimes accompanied by a few tiny gastroliths (*Figure 4A*, *Figure 4—figure supplement 1A, C, D*). The seed remains preserved in currently known *Jeholornis* specimens most closely resemble the condition in frugivores, being completely intact and sparsely dispersed (*Figure 3A–C*) compared to the gastroliths preserved in other individuals (*Figure 3D*; *O'Connor, 2019*; *O'Connor et al., 2018*; *O'Connor and Zhou, 2020*).

## Discussion

Digital reconstruction of an exceptionally well-preserved new specimen of the early-diverging bird *Jeholornis* reveals a plesiomorphic, diapsid skull, sharing numerous features with non-avian theropods. These features include a complete postorbital bar, unreduced squamosal, and unmodified palate (*Hu et al., 2020b*, *Hu et al., 2019*; *Rauhut et al., 2018*), reinforcing evidence for an early-diverging phylogenetic position among birds (*Wang et al., 2018*; *Zhou and Zhang, 2002*). Nevertheless, compared to *Archaeopteryx* (*Rauhut, 2014*; *Rauhut et al., 2018*), *Jeholornis* also possesses clear diet-related specialisations of the rostrum including partial fusion of the premaxillae and a strongly reduced dentition.

Our GMM analyses reveal that the mandibular and cranial shapes of *Jeholornis* and *Sapeornis* are distinct from those of seed-cracking granivorous birds, consistent with earlier assumptions that the delicate, vestigial dentary teeth of *Jeholornis* would be too prone to damage if used to de-husk hard foods (*Ksepka et al., 2019*; *Mayr et al., 2020*; *O'Connor et al., 2018*; *O'Connor and Zhou, 2020*), and contrary to previous claims that the reportedly 'deep' mandible of *Jeholornis* is suitable for such a behaviour (*Zhou and Zhang, 2002*). Although their mandibular and cranial shapes occupy the morphospace in which several diets overlap, including frugivory and gastric seed-grinding granivory, these diets can be distinguished through comparing the condition of ingested remains in the alimentary tract in modern birds.

Known stomach contents preserved in *Jeholornis* take two forms in different fossil specimens, which include: (1) individuals with sparsely distributed and entirely intact seeds (*Figure 3A–C*; *Zhou and Zhang, 2002*), and (2) those with a relatively small concentration of gastroliths without any seed remains (*Figure 3D*; *O'Connor et al., 2018*). Our comparisons with modern birds indicate that the first group of *Jeholornis* individuals ingested fleshy propagules (fruit consumption), rather than consuming seeds for nutrient extraction (destructive seed consumption). We cannot interpret the presence of gastroliths in the second group of individuals, because gastroliths are widespread in extant birds with a wide range of diets for example insectivory, granivory, and frugivory (*Gionfriddo and Best, 1996*; *O'Connor, 2019*; *Piersma et al., 1993*; *Wings, 2007*), making it impossible to infer diet from this evidence alone. Crucially, no *Jeholornis* specimen preserves seeds and gastroliths together (*O'Connor, 2019*; *O'Connor and Zhou, 2020*) (and preserved seeds within *Jeholornis* are not abraded), which would be required as evidence for seed-grinding granivory.

Variation in alimentary contents among individuals of *Jeholornis* are best interpreted as evidence of seasonal variation in diet, or potentially other intraspecific variation in diet (*O'Connor, 2019*; *O'Connor et al., 2018*). Though the influence of preservational biases cannot be completely

excluded yet, the recurring occurrence of specific sets of stomach contents among individuals suggests that these reflect habitual rather than exceptional dietary variation. It is possible that *Jeholornis* consumed fleshy propagules during the seasons in which such food sources were available, but fed on other food sources during other seasons, which is also consistent with the seasonal climate of the western Liaoning region during the Early Cretaceous (*Ding et al., 2006*). However, we currently lack strong evidence of what diet items were consumed by *Jeholornis* in addition to fruits. Mandibular and cranial shape excludes *Jeholornis* from being having a probing/piscivorous diet, and is consistent with omnivory, grabbing/pecking for invertebrates, or processing foliage (using the gastric mill). Seasonal dietary shifts are widely known in modern birds that feed on fruits as a substantive part of their diet such as Ruffed Grouse (*Bonasa umbellus*) and Hoatzin (*Opisthocomus hoazin*) (*Billerman et al., 2020*), since plants usually bear fruits only in certain seasons rather than throughout the year (*Corlett, 1998*; *Howe, 1986*; *Jordano, 2014*; *Wilman et al., 2014*). Our findings suggest that the dietary flexibility of fruit consumption may be traced back to the earliest stages of bird evolution.

The evidence for fruit consumption in *Jeholornis* demonstrates that early birds with seeds preserved in the abdominal area cannot be identified as granivores without further evidence from cranial morphology, or the co-occurrence of abraded or fragmented seeds with gastroliths. It was recently suggested that *Sapeornis*, *Eogranivora*, and even some enantiornithines may have consumed fruits (*Ksepka et al., 2019*; *Mayr et al., 2020*). However, the evidence for this remains equivocal. *Sapeornis* and *Eogranivora* preserve apparently whole seeds in the crop, but only gastroliths in the abdominal area (*Zheng et al., 2018*; *Zheng et al., 2011*), consistent with both seed-grinding granivory and passerine-like seed-cracking (*Figure 4C, D*). Therefore, *Jeholornis* is so far the only Mesozoic bird that provides strong evidence of fruit consumption. However, this should not be taken as evidence that fruit consumption was rare. Direct evidence on diet in fossil birds is rare and preserved gut contents are limited to just a few individuals from a small number of Early Cretaceous fossil deposits in China and Europe (*O'Connor, 2019*; *Miller and Pittman, 2021*). Given this low level of current knowledge, evidence for fruit consumption in *Jeholornis* is important in demonstrating for the first time that at least some early birds ate fruits.

Flight-related anatomical specialisations suggest that *Jeholornis* was a competent flier in spite of its early-diverging phylogenetic position (*Pei et al., 2020*; *O'Connor et al., 2013*; *Zheng et al., 2020*; *Zhou and Zhang, 2002*). Although flight is not an exclusive adaptation to fruit consumption, compared to non-volant animals, flight allows birds and bats to more easily obtain patchily distributed but energy-rich food sources in difficult to access and widely dispersed locations, including fruits (*Benson et al., 2018b*, *Benson et al., 2014a*; *Maurer, 1998*), and may in part explain the high prevalence of fruit consumption (and especially the consumption of small fruits such as berries) among extant birds compared to most other tetrapod groups (*Tiffney, 2004*; *Tiffney, 1992*).

Although true fruits are only present in angiosperms, seed ferns, and gymnosperms evolved functionally analogous fleshy-coated propagules such as arils and other fleshy accessory tissues much earlier (*Tiffney, 1986*; *Tiffney, 2004*; *Herrera, 1989*; *Lovisetto et al., 2012*; *Contreras et al., 2017*; *Herendeen et al., 2017*). Such structures represent specialisations for animal-mediated seed dispersal. Early fruit-producing angiosperms were present by the Early Cretaceous (*Eriksson et al., 2000b*), alongside multiple groups of gymnosperms with fleshy propagules including cycadales, ginkgoales, and gnetales (*Tiffney, 1986*; *Tiffney, 2004*; *Wu, 1999*) – which are also present in Jehol Biota (*Leng and Friis, 2003*; *Sun et al., 2001*). The alimentary contents preserved in *Jeholornis* were preliminarily described as ginkgo-like seeds (*Zhou and Wu, 2006*) and more likely to be gymnospermous due to their relatively large sizes, but have not been confidently identified with detailed comparisons with all the potential Early Cretaceous fruits/arils. In addition, although the poor preservation of these ingested seeds prevents any detailed taxonomic identification, three morphotypes have been grouped in previous studies based on size and shape: morphotype-1 in smaller size with a circular shape and curved striations, morphotype-2 in larger size with an oval shape, and morphotype-3 in similar size to morphotype-1 but with a strongly tapered pole (*O'Connor et al., 2018*). Therefore, considering that early birds such as those from the Jehol Biota would encounter both gymnosperms and angiosperms, we suggest that during the origin of fruit consumption among birds, early frugivorous birds were likely to be opportunistic and targeted fleshy propagules from both groups, rather than being 'gymnosperm specialists'.

Given the importance of frugivorous birds today as agents of seed dispersal (*Pejchar et al., 2008*; *Sekercioglu, 2006*; *Tiffney, 2004*), the early occurrence of fruit consumption in birds may signify the origin of an important component of modern-like biotic dispersal systems, providing new opportunities for co-evolutionary mutualisms, though future research is expected to provide solid confirmation for this hypothesis. The occurrence of specialised seed dispersal by animals during the Early Cretaceous has previously been proposed indirectly, based on the presence of aril-producing gymnosperms and early fruit-producing angiosperms (*Eriksson, 2008*; *Eriksson et al., 2000a*). However, the identification of these fruit eaters has been uncertain and fruit consumption was almost unmentioned in the recent review of early bird diets, owing to the lack of available evidence (*Miller and Pittman, 2021*). Evidence for fruit consumption in *Jeholornis* provides direct evidence of fruit consumption in early birds, long before the origin of the bird crown-group. This provides an important indication of the possibility that birds were recruited by plants for seed dispersal very early in their evolutionary history, during the Early Cretaceous.

Fossil birds have low preservation potential and are known primarily from sites of exceptional preservation. Outside of the Jehol Biota, the fossil record of early birds is poorly sampled, both in space and time. However, evidence from less complete fossil remains suggests that birds had a wide geographic distribution by the Early Cretaceous (*Chiappe and Witmer, 2002*; *Close et al., 2009*), suggesting a 'hidden' taxonomic, and most likely ecological, diversity of Mesozoic birds. Diversification of birds therefore may explain, at least in part, the evolutionary expansion of fruit abundance, especially angiosperm fruits, that occurred through the Cretaceous (*Eriksson, 2008*; *Eriksson et al., 2000a*). Direct evidence for the diet of extinct species is rare. However, evidence in *Jeholornis* indicates the potential for at least opportunistic fruit consumption among early birds in general. It therefore increases support for the hypothesis that bird–plant interactions are likely to have played at least some role in the Cretaceous Terrestrial Revolution (*Tiffney, 2004*). Specifically, the occurrence of fruit consumption in one of the earliest-diverging bird lineages raises the possibility of synergistic evolutionary influences, with birds enabling seed dispersal for plants, and obtaining a rich energy resource in return (*Muller-Landau and Hardesty, 2005*; *Dennis, 2007*; *Jordano, 2014*; *Carlo and Morales, 2016*; *Carlo et al., 2022*). New discoveries and comparative analyses are required to test this hypothesis, by deeper insights into the ecologies of early bird species, and the potential role of the birds during the transition from gymnosperm- to angiosperm-dominated floras.

## Materials and methods
### Taxonomy of *Jeholornis* STM 3–8

*Jeholornis* STM 3–8 was collected from the Jiufotang Formation (~120 Ma) (*He et al., 2004*) at the Dapingfang locality in Chaoyang, Liaoning province, preserving a complete and mostly articulated skull, and a few postcranial elements including the vertebral column, the pelvic girdle and fragmentary hindlimbs. This new specimen is tentatively assigned to *Jeholornis prima* based on the presence of the following features: relatively robust mandible with three rostrally restricted teeth; edentulous and robust premaxilla; maxilla lacking teeth in the caudal portion; long bony tail consisting of more than 20 caudal vertebrates. This specimen could be distinguished from *Jeholornis palmapenis* by its flattened dorsal margin of ilium, compared to the strongly convex condition in *J. palmapenis* (*O'Connor et al., 2013*). The validity of another recently reported jeholornithiformes, *Kompsornis longicaudus* (*Wang et al., 2020*) needs more discussions since only one specimen is used to erect it, while no detailed comparisons have been done to the numerous specimens which have been assigned to *Jeholornis* before. In addition, the parts bearing key features listed in *Wang et al., 2020* such as pectoral girdle and sternum, are not preserved in STM 3–8. However, some characters such as the relatively pointed rostral tip of the mandible of *Kompsornis* still tentatively indicate that STM 3–8 may be distinguished different from it.

### CT scans and digital reconstructions

Microtomographic measurements of *Jeholornis* STM 3–8 were performed using the Imaging and Medical Beamline (IMBL) at the Australian Nuclear Science and Technology Organisation's (ANSTO) Australian Synchrotron, Melbourne, Australia. For this investigation, acquisition parameters included a pixel size of 16.9 × 16.9 µm, monochromatic beam energy of 70 keV, a sample-to-detector distance

of 200 mm. As the height of the specimen exceeded the detector field-of-view, the specimen was aligned axially relative to the beam and imaged using seven consecutive scans. The raw 16-bit radiographic series were normalised relative to the beam calibration files and stitched. Reconstruction of the 3D dataset was achieved by the filtered-back projection method using the CSIRO's X-TRACT (*Gureyev et al., 2011*).

The 3D reconstructions (*Figure 1A, B*) and the fixing of 3D models (*Figure 1C, D*) were created and completed with the software Mimics and 3-matic (version 16.1). The mandible model of *Jeholornis* STM 3–8 was reconstructed for the GMM analysis (*Figure 1H*) by the following steps: the crashed left splenial was replaced by the mirrored right splenial; the breakage through the left dentary and surangular was joined together; second left dentary tooth was replaced by the mirrored right counterpart with better preservation; all the left dentary teeth were slightly relocated according to the morphology of the alveoli; the fixed left mandible was then mirrored to create the right half; the two sides were joined together, with the angle between them determined by the width of the braincase. The 3D models of the cranial elements of *Jeholornis* STM 3–8 were reassembled (*Figure 1C, D*) by the following steps: all the left elements with better preservation were mirrored to create the right half, except for the pterygoid, for which the better-preserved right one was used as the reference; all the elements were relatively relocated to build a complete skull according to their articulations and anatomical geometry. Since most elements are only slightly dislocated with the articulations/articulation facets preserved, this reassembled model is largely reliable, with the location of the preorbital ossifications being the highest uncertainty. The reassembled cranial model was then used as the reference for the 2D reconstruction of the *Jeholornis* skull in lateral and ventral views (*Figure 1E, F*). However, since the braincase is too flattened to be used as the reference for 3D retrodeformation, it was omitted in *Figure 1C* and reconstructed according to its common shape in early birds in *Figure 1E*. The ectopterygoid is not preserved but suspected to exist as discussed in the Cranial Anatomy part, therefore it was reconstructed according to the shape of this element among other stem birds for example *Archaeopteryx* and *Sapeornis* (*Elzanowski and Wellnhofer, 1996*; *Hu et al., 2019*).

## GMM analyses

The dataset incorporates *Jeholornis* and 160 extant bird species representing 111 families and 36 orders in our 3D mandible analysis, with additional Mesozoic theropods in 2D skull analysis including: *Sinornithosaurus* (Dromaeosauridae) (*Xu and Wu, 2001*), *Linheraptor* (Dromaeosauridae) (*Xu et al., 2015*), *Dilong* (Tyrannosauroidea) (*Xu et al., 2004*), *Archaeopteryx* (non-Ornithothoraces Aves) (*Rauhut, 2014*), *Sapeornis* (non-Ornithothoraces Aves) (*Hu et al., 2019*), *Pengornis* (Enantiornithes) (*O'Connor and Chiappe, 2011*), and *Ichthyornis* (Ornithuromorpha) (*Field et al., 2018*). We note that the 2D cranial reconstruction of *Pengornis* is less reliable among those Mesozoic samples due to the comparatively poor preservation, but we incorporate it here as it is currently the best representative enantiornithine.

One anatomical landmark and four curves (semi-landmarks) were placed in each mandible in 3D, and five anatomical landmarks and five curves were placed in each cranium in 2D, using Avizo Lite (version 9.2.0). Landmark definitions and descriptions are modified from *Bjarnason and Benson, 2021* (details see *Figure 2—figure supplement 1* and *Figure 2—source data 1*, *Figure 2—source data 2*). All the digital landmarks and semi-landmarks were imported into R (version 3.6.0) for further analyses. A GPA was performed on all landmarks using the gpagen() function from the R package 'geomorph', to rotate, translate, and scale landmark configurations to unit centroid size (*Adams et al., 2013*; *Goodall, 1991*; *Rohlf and Slice, 1990*). To visualise the multivariate ordination of the aligned Procrustes coordinates, a PCA was performed afterward using plotTangentSpace() from 'geomorph'. The shape variations of both 3D mandible and 2D skull along different PC axes were visualised using plotRefToTarget() from 'geomorph'.

The ecological information including diet categories and foraging strategies of modern birds were modified from *Wilman et al., 2014*. The diets of birds were originally assigned to five categories: (1) Plant and Seeds; (2) Fruits and Nectar; (3) Invertebrates; (4) Vertebrates and Fish and Carrion; and (5) Omnivore (*Wilman et al., 2014*). Based on our focal goal and information from *Birds of the World* (BOW) (*Billerman et al., 2020*), those categories were either split or merged to form five new categories in our main analysis: (1) Seed-crackers (parrots): Psittaciformes; (2) Seed-crackers (passerines): mostly finches including Fringillidae, Thraupidae, and Sylviidae, and some other granivorous

passerines; (3) Seed-grinders: galliforms and members of Columbidae, Anatidae, Alaudidae, Odontophoridae, Tinamidae, Pedionomidae, and Pteroclidae; (4) Fruit eaters: members of Paradisaeidae, Phasianidae, Calyptomenidae, Capitonidae, Coliidae, Musophagidae, Cracidae, Megalaimidae, Opisthocomidae, Pipridae, Psophiidae, Columbidae, Ramphastidae, Cotingidae, Tityridae, and Trogonidae, as well as the frugivorous parrot *P. fulgidus* (Pesquet's Parrot); (5) Other diets (such as other herbivores, carnivores, and omnivores). Among them, the diets of three modern species were modified according to BOW (*Billerman et al., 2020*): *Anas discors* modified to be 'Seed-grinders' from 'Omnivore', which is also consistent with other anatids; *Psittacus erithacus* modified to be 'Seed-crackers (parrots)' from 'Fruits and Nectar' since it has the ability and occasionally does crack and eat seeds; *P. torquatus* modified to be 'Seed-grinders' from 'Omnivore', since its diet includes 30% of seeds and its complexity is discussed in Results. The modified diet categories were used to group the samples in the PCA results of the main analysis (*Figure 2*).

The category 'Other diets' was further split to eight categories in our supplemental analysis primarily based on the information from *Wilman et al., 2014* and *Tobias et al., 2022*: (1) Probing for invertebrates; (2) Grabbing/pecking for invertebrates; (3) Piscivores: including taxa who have a mixed fish/cephalopod diet; (4) Animal-dominated omnivores: including taxa who have >65% animals in diet; (5) Carnivores; (6) Nectarivores; (7) Omnivores: including taxa who have approximately even split of animals and plants in diet; (8) Plant-dominated omnivores: including taxa who have>65% plants in diet.

## Detailed descriptions of the alimentary contents in modern birds

1. Frugivores: *M. comrii* (Curl-crested manucode, *Figure 4A*, *Figure 4—figure supplement 1A*) is a specialised fruit eater (*Billerman et al., 2020*). Several whole fruits are revealed along the alimentary tract of our sample, each including four intact, unabraded seeds in a regular configuration, as well as another kind of disc-shaped seeds, and no gastroliths are preserved (*Figure 4A*; *Figure 4—figure supplement 1A*). Another frugivore *B. garrulus* (Bohemian waxwing, *Figure 4—figure supplement 1C, D*) was also sampled, and the same situation of the contents is revealed as in *M. comrii*. All the seeds preserved through its alimentary tract including crop, stomach and intestines are intact, and more sparsely located than in the seed-grinders and seed-crackers that we sampled.

2. Seed-cracking parrots: *C. carolinensis* (Carolina parakeet, *Figure 4E*), a parrot, is a specialised seed-cracker using beak to de-husk the seeds (*Billerman et al., 2020*). The alimentary tract of this sample contains a proportionally small bolus of highly fragmented seeds with original shapes impossible to determine, and very few small and sparse stones.

3. Seed-cracking passerines: *G. fuliginosa* (Small ground-finch, *Figure 4D*, *Figure 4—figure supplement 1E*) is half a seed-cracker and half a seed-grinder, and has a diet mostly consisting of small seeds (*Billerman et al., 2020*). The crop contents of this sample consist of seeds with almost intact configuration, whereas those in the stomach are highly fragmentary along with lots of large gastroliths. We then sampled another seed-cracking passerine *C. lapponicus* (Lapland longspur, *Figure 4—figure supplement 1F*), and found the same situation of the contents as in *G. fuliginosa*.

4. Seed-grinding granivores: *E. migratorius* (Passenger pigeon, *Figure 4C*, *Figure 4—figure supplement 1B*), a seed-specialist pigeon, is a seed-grinder that entirely uses gastroliths to crack the seeds (*Billerman et al., 2020*). Its crop contains numerous, well-defined and intact seeds, whereas seeds are highly fragmented in the stomach, similar to those in *C. carolinensis* and *G. fuliginosa*, together with two large, round gastroliths. Another representative, *P. torquatus* (Plains-wanderer, *Figure 4B*, *Figure 4—figure supplement 2A–D*) is a general, small-sized seed-grinder. The seeds preserved in the alimentary tract of *P. torquatus* are comparatively more intact than those in other seed specialists such as parrots, pigeons, and finches, but many seeds show partial breakages and the gastroliths they contained are much smaller. This indicates that *P. torquatus* might utilise another strategy of abrasion to digest the seeds rather than entirely fragmentation. To test this interpretation, we sampled another seed generalist, *T. rumicivorus* (Least seedsnipe, *Figure 4—figure supplement 2E, F*). The seed remains are in the same condition as in *P. torquatus* – not fragmentary but abraded with partial breakages, along with small gastroliths, confirming the strategy used by those general seed-grinders.

## Acknowledgements

We acknowledge Dahan Li and Wei Gao (Institute of Vertebrate Paleontology and Paleoanthropology, Chinese Academy of Sciences) for specimen preparation and photography; Matt White and Gabriele Sansalone (University of New England, Australia) for help with CT scanning and discussions; Anton Maksimenko for technical assistance with the synchrotron imaging; Andrew Orkney and Duhita Naware (University of Oxford, UK) for discussions; and Zhixin Han and Yifan Wang for ecological reconstruction illustrations. This research is part of a project that has received funding from the European Union's Horizon 2020 research and innovation programme under the Marie Skłodowska-Curie grant agreement No 101024572. It is also supported by a Postdoctoral Research Fellowship from the University of New England; Anne Sleep Award from the Linnean Society of London; project ZR2020MD026 supported by Shandong Provincial Natural Science Foundation, China; Linyi Key Research and Development Project 2020ZX028; and the National Natural Science Foundation of China grant 42288201, 41402017, and 42002016. Access to the Australian Synchrotron's Imaging and Medical Beamline was granted under proposal M13126.

## Additional information

### Funding

| Funder | Grant reference number | Author |
| --- | --- | --- |
| University of New England | Postdoctoral Research Fellowship | Han Hu |
| Linnean Society of London | Anne Sleep Award | Han Hu |
| Shandong Provincial Natural Science Foundation | ZR2020MD026 | Yan Wang |
| Linyi Key Research and Development Project | 2020ZX028 | Yan Wang |
| National Natural Science Foundation of China | 42288201 | Zhonghe Zhou |
| National Natural Science Foundation of China | 41402017 | Yan Wang |
| Australian Synchrotron's Imaging and Medical Beamline | M13126 | Han Hu |
| National Natural Science Foundation of China | 42002016 | Yan Wang |
| European Union's Horizon 2020 research and innovation programme under the Marie Skłodowska-Curie grant agreement | No 101024572 | Han Hu |

The funders had no role in study design, data collection, and interpretation, or the decision to submit the work for publication.

### Author contributions

Han Hu, Conceptualization, Data curation, Formal analysis, Funding acquisition, Investigation, Methodology, Writing – original draft, Project administration, Writing – review and editing; Yan Wang, Conceptualization, Resources, Funding acquisition, Investigation, Writing – review and editing; Paul G McDonald, Stephen Wroe, Supervision, Investigation, Writing – review and editing; Jingmai K O'Connor, Investigation, Writing – original draft, Writing – review and editing; Alexander Bjarnason, Data curation, Formal analysis, Investigation; Joseph J Bevitt, Resources, Funding acquisition, Investigation; Xuwei Yin, Xiaoting Zheng, Resources, Investigation; Zhonghe Zhou, Conceptualization,

Resources, Supervision, Funding acquisition, Investigation, Writing – review and editing; Roger BJ Benson, Conceptualization, Formal analysis, Supervision, Investigation, Methodology, Writing – original draft, Writing – review and editing

### Author ORCIDs
Han Hu ⬥ http://orcid.org/0000-0001-5926-7306

### Decision letter and Author response
Decision letter https://doi.org/10.7554/eLife.74751.sa1
Author response https://doi.org/10.7554/eLife.74751.sa2

## Additional files

### Supplementary files
• Transparent reporting form

### Data availability
The new specimen reported here (*Jeholornis* STM 3-8) is housed and available for future researchers to check at Shandong Tianyu Museum of Nature, China. The original CT scanning slices and segmented STL files of *Jeholornis* STM 3-8 and involved modern birds, and the alimentary contents of selected modern birds are available in Morphosource (https://www.morphosource.org/projects/0000C1212; https://www.morphosource.org/projects/00000C420; https://www.morphosource.org/projects/0000C1080). Other data that support this study are available in Figshare (DOI: 10.6084/m9.figshare.13217672), including the rotating videos of the original/reassembled cranial 3D models of *Jeholornis* STM 3-8 and the 3D models of the alimentary contents of selected modern birds, and the landmark data and taxa lists used in GMM analyses. Further information and requests for resources should be directed to and will be fulfilled by the Lead Contacts, Yan Wang (wangyan6696@lyu.edu.cn) and Han Hu (han.hu@earth.ox.ac.uk).

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
