## [Editor Report]

This article provides important new information on the ecology and morphology of a phylogenetically and temporally interesting early avialan. The work has important implications that should stimulate future research on Mesozoic bird-plant interactions.

---

## [Decision Letter]

**Decision letter after peer review:**

Thank you for submitting your article "Earliest evidence for frugivory and seed dispersal by birds" for consideration by *eLife*. Your article has been reviewed by three peer reviewers, and the evaluation has been overseen by a Reviewing Editor and Christian Rutz as the Senior Editor. The following individuals involved in the review of your submission have agreed to reveal their identity: Nathan Jud (Reviewer #2); Chris Torres (Reviewer #3).

The reviewers have discussed their reviews with one another, and the Senior Editor has drafted this decision letter to help you prepare a revised submission.

Essential revisions:

The reviewers agreed that this is an exceptional specimen, but have raised reservations about the analyses and inferences presented. In light of their feedback, we would like to request the following essential revisions:

1. Please critically evaluate the alternative hypothesis that *Jeholornis* is neither frugivorous nor granivorous and that the observed gut contents instead represent facultative frugivory. In our view, addressing this point requires additional analyses, rather than just text revision.

2. The (co-)evolutionary implications of this discovery remain unclear as presented, and need to be explored further. The seeds are perhaps more likely gymnospermous than angiospermous. There are multiple origins of fleshy accessory tissues in gymnosperms, suggesting that plants were recruiting vertebrate seed dispersers repeatedly, and although crown birds may not be ancestrally frugivorous, the multiple origins of frugivory in crown birds suggests dietary lability. The abundance of animal-dispersed plants, and the frequency with which frugivory evolved in crown birds means that the appearance of facultative, opportunistic, or seasonal frugivory in stem birds is perhaps not particularly surprising.

3. What is the evidence that seed dispersal by frugivorous birds enhances diversification rates through increased speciation rates or decreased extinction rates in either plants or birds? Please cite relevant studies that evaluated this hypothesis.

4. The treatment of the plant material seems weak, as the seeds are not described and the brief mention of gymnospermous accessory tissues masks complex plant-animal interactions related to seed dispersal among non-angiosperms. Please improve documentation and analyses.

5. Please address the points raised in the reviewers' full reports, which are appended below.

6. Given the concerns raised, there was a feeling that claims were worded too strongly, and that language should be toned down throughout.

Please note that, in light of the fact that the study's key claim currently seems insufficiently supported, we will send your revised manuscript out for re-evaluation and that eventual acceptance is not guaranteed.

*Reviewer #1 (Recommendations for the authors):*

Our main comment is related to experimental design as the possibility that *Jeholornis* was not a specialist granivore or frugivore was not evaluated to the same extent as granivory and frugivory in your study. This third hypothesis would be important to address using multiple methods simultaneously. This is supported by the PCA data where there is overlap between the granivory and frugivory data points and the 'other diet' data points.

The idea of a bird-plant link in the Early Cretaceous is interesting and could be spelled out more, along with further consideration of the uncertainties revealed in key studies e.g., Hulme 2002.

Hulme, P.E., 2002. Seed-eaters: Seed Dispersal, Destruction and Demography. Seed dispersal and frugivory: Ecology, evolution, and conservation, p.257.

However, this is secondary to our main comment above which is related to experimental design and the data incompletely supporting the conclusions proposed.

We suggest two options to consider [### Note from the Senior Editor: our clear preference is Option 2 ###]:

1) Narrow the scope of the conclusions to those which can be well supported by the data available i.e., a study showing that, for the seed components of the diet of *Jeholornis*, whether granivory or frugivory is most supported.

2) Alternatively, we suggest to undertake a more holistic diet analysis based on multiple lines of evidence that tests hypothesis 3 and then using the results to inform the conclusions (which may or may not be related to granivory or frugivory).

We hope you find the comments provided helpful.

*Reviewer #2 (Recommendations for the authors):*

As far as I can tell, the descriptions and comparisons of skull morphology are detailed and well done. The argument that *Jeholornis* was not a seed cracker is strong, and the interpretation of *Jeholornis* as at least seasonally frugivorous is well-reasoned. It does nicely show that volant birds were recruited for endozoochory early. It will be a pleasure to see this published. Nonetheless, I do have some minor questions and concerns/recommendations about the botany and co-evolutionary implications that I think would make the manuscript stronger, along with some minor suggestions regarding the writing clarity that can be adopted at the authors' discretion.

Line 60. Using plesiomorphic as an adverb strikes me as odd. It seems to me the meaning the sentence is unchanged if the word plesiomorphically is omitted and instead it reads "retaining an elongate bony tail"

Line 64. In what way is *Jeholornis* the most representative jeholornithiform? Are other jeholornithiforms somehow less representative?

Line 75. Consider substituting or adding "seed predation" alongside granivory. This widely used term emphasizes the ecological difference between feeding on seeds vs. other forms of herbivory as it kills a "whole plant."

Line 77. Consider changing to "mutualistic co-evolutionary influence," or perhaps "co-diversification" is what you mean; however, see my comment about lines 342-343 below.

Like 86. "Complete" or "precise" might be a better word choice than "accurate." Are previous reconstructions inaccurate? Perhaps they are just incomplete or poorly resolved, but not inaccurate.

Line 95. Because of its phylogenetic position… Because modifies a verb, in this case "studied" "Due to" means "caused by". Or, to put it another way, *Jeholornis* has been frequently studied because of its phylogenetic position.

Line 130. I suggest "indicates that rostral fusion evolved earlier (or deeper in bird phylogeny) than previously thought." I also suggest that the authors point out what other characters it now precedes to emphasize the point that paleontology reveals the order of character changes along branches.

Line 225: I don't think the parentheses are necessary.

Lines 260-263. I encourage the authors to describe the seeds, or at least to summarize the morphological diversity, number of types, (and perhaps their organization) if they were all previously described in O'Connor et al. (2018). What are their morphology, dimensions, and size variation? Do all of them have longitudinal striations? Based on the figures, these seeds appear to be quite large for Cretaceous angiosperms (Tiffney 1984; Eriksson et al. 2000; 2008). Is it more likely that these are seeds of a gymnospermous plant? I suspect so.

Lines 274-277. The hypothesis about seasonal frugivory in *Jeholornis* is both reasonable and interesting. Some indication of warm climate with seasonality of precipitation is supported elsewhere in the Jehol biota by Ding et al.'s (2016) analysis of fossil woods. It may be worth citing that work.

Lines 305-315. Extant gymnosperms that use endozoochory for seed dispersal use a variety of accessory tissues, not just arils. Ginkgo, cycads, and some extinct groups have a fleshy sarcotesta (outer seed coat). An aril is a fleshy appendage of the seed coat or the funiculus. Some gymnosperms have fleshy subtending bracts (Ephedra), and some have fleshy accessory tissues such as bracts (Ephedra), cones (Juniperus), receptacle and epimatium (Podocarpaceae), or fleshy cones with numerous seeds (Juniperus). See Tiffney (1986, 2004); Herrera (1989); Lovisetto et al. (2012); Contreras et al. (2017); Nigris et al. (2021). I suggest Herrera (1989) as an earlier citation than Herendeen et al. (2017) for line 306.

Line 318. Why small? What does small mean? Small enough to fit in their mouth?

Line 321. I suggest "early and repeated" instead of "ancient". It appears to be clear that there are multiple origins of frugivory/omnivory within crown group birds, and it is not clear that the MRCA of crown-group birds was frugivorus (Felice et al. 2019). This lability in bird diet and repeated origin of frugivory might actually better support the authors' argument that diffuse coevolution between angiosperms and frugivorous birds contributed the KTR. There is now good evidence that volant birds have been repeatedly recruited by plants as endozoochorous seed dispeserers, even prior to the evolution of the MRCA of extant birds.

Line 342-343. Can you provide a citation linking dispersal mode to diversification rate? Eriksson and Bremer (1992) in the journal 'Evolution' found no relationship between dispersal mode and diversification rate in angiosperms, although Carlo and Morales (2015) in the journal 'Ecology' showed evidence that frugivorous birds increase plant α diversity.

---

## [Author Response]

Essential revisions:The reviewers agreed that this is an exceptional specimen, but have raised reservations about the analyses and inferences presented. In light of their feedback, we would like to request the following essential revisions:1. Please critically evaluate the alternative hypothesis that *Jeholornis* is neither frugivorous nor granivorous and that the observed gut contents instead represent facultative frugivory. In our view, addressing this point requires additional analyses, rather than just text revision.

Additional analyses and dissection of the morphometric results have been added, and details of them are provided in the following reply to Reviewer 1. These additional analyses delimit among a larger number of dietary categories than our original analyses, which we believe is what Referee 1 required.

Note that our original manuscript did claim facultative frugivory, and we had consistently used the terms ‘seasonal frugivory’ or ‘partial frugivory’ in our initial submission. Also that we suggested explicitly that *Jeholornis* may have taken other diet items including e.g. insectivory. Nevertheless, we have attempted to clarify this further in the revised manuscript (also see below in the reply to Reviewer 1). In particular, we now refer more often to ‘fruit consumption’, describing the behaviour of taking fruit instead of a dietary class. Note that this is not so different to many extant birds, which show different levels of frugivory, mainly taking fruit (including many temperate birds taking berries) or only as a proportion of the diet rather than their strict diet class.

2. The (co-)evolutionary implications of this discovery remain unclear as presented, and need to be explored further. The seeds are perhaps more likely gymnospermous than angiospermous. There are multiple origins of fleshy accessory tissues in gymnosperms, suggesting that plants were recruiting vertebrate seed dispersers repeatedly, and although crown birds may not be ancestrally frugivorous, the multiple origins of frugivory in crown birds suggests dietary lability. The abundance of animal-dispersed plants, and the frequency with which frugivory evolved in crown birds means that the appearance of facultative, opportunistic, or seasonal frugivory in stem birds is perhaps not particularly surprising.

We think this point may be separated to three parts:

1) By providing direct evidence of fruit-consumption in early stem birds, we provided the mechanism for the bird-plant co-evolutionary mutualism. Our study provides the first direct clear evidence of habitual fruit-consumption in an early-diverging bird outside the crown group, and this is the importance of our findings. Direct evidence of co-evolution itself is almost impossible to preserve in fossils. Therefore, even in our initial submission, we opted to tone down the relevant statements rather than making them too strong and specific. We generally restricted our discussions to the first step to this topic – the existence of the mechanism not the actual, direct evidence of this mutualism.

We think that the evidence of fruit-consumption in stem birds that we provided here is important. The referee summary suggests that this might be thought to be likely, even in absence of evidence. But that is a hypothesis and is contingent on various factors, in particular that crown-group birds are a good model for stem birds, in spite of various anatomical and functional differences. Our data provides a test of this hypothesis, which is important beyond the supposition of what might have been likely or unlikely in absence of evidence. It is worth noting that fruit, and other animal-dispersed reproductive structures (e.g. the fleshy diaspores of gymnosperms) were probably not as abundant in the Early Cretaceous as they are today (e.g. see works by Ericksson on fruit evolution). Therefore, the world was not the same back then as it is today, and it is not strictly correct to say that the high frequency of fruit-consumption by birds today makes the occurrence of fruit-consumption in the Early Cretaceous unsurprising. In fact, the importance and frequency of this interaction today is precisely what makes it so important to find direct evidence on the question of whether it happened at all in the Early Cretaceous.

2) This point is also mentioned in the “Line 321” comment from Reviewer 2. The lability in bird diet and multiple origins of frugivory in crown birds actually supports our argument that although we cannot decide the gymnosperm or angiosperm affiliation of the gut content in *Jeholornis* yet, considering that early birds such as those from the Jehol Biota would encounter both gymnosperms and angiosperms, early frugivorous birds were likely to be opportunistic and targeted small, fleshy propagules from both groups, rather than being ‘gymnosperm specialists’.

3) As we stated in the Discussion part: “The occurrence of specialised seed-dispersal by animals during the Early Cretaceous has previously been proposed indirectly, based on the presence of aril-producing gymnosperm and early fruit-producing angiosperms (Eriksson, 2008; Eriksson et al., 2000a). However, the identification of these frugivores has been uncertain and frugivory was almost unmentioned in the recent review of early bird diets (Miller and Pittman, 2021). Evidence for at least seasonal frugivory in *Jeholornis* provides direct evidence of frugivores and thus indicated highly likely seed-dispersal by animals during the Early Cretaceous for the first time.” We evidenced a long-standing speculation in this area that many researchers are interested – we think this is how important scientific works are made. It is like a “missing link” has been supposed to exist between birds and non-avian reptiles for a long time, but the discovery of *Archaeopteryx* evidenced it and showed the world how it looks – maybe also not “surprising” but no doubt important and inspiring.

Eriksson O. 2008. Evolution of seed size and biotic seed dispersal in Angiosperms: Paleoecological and Neoecological Evidence. *Int J Plant Sci* 169:863–870. doi:10.1086/589888

Eriksson O, Friis EM, Löfgren. 2000. Seed size, fruit size, and dispersal systems in Angiosperms from the Early Cretaceous to the Late Tertiary. *Am Nat* 156:47–58. doi:10.1086/303367

Miller CV, Pittman M. 2021. The diet of early birds based on modern and fossil evidence and a new framework for its reconstruction. *Biol Rev*. doi:10.1111/brv.12743

3. What is the evidence that seed dispersal by frugivorous birds enhances diversification rates through increased speciation rates or decreased extinction rates in either plants or birds? Please cite relevant studies that evaluated this hypothesis.

Reply to this point are provided in the following reply to Reviewer 2’s comment of “Line 342-343”.

4. The treatment of the plant material seems weak, as the seeds are not described and the brief mention of gymnospermous accessory tissues masks complex plant-animal interactions related to seed dispersal among non-angiosperms. Please improve documentation and analyses.

Reply to this point are provided in the following reply to Reviewer 2’s comment of “Line 260-263”.

5. Please address the points raised in the reviewers' full reports, which are appended below.

Every point in the reviewers’ reports is addressed point to point.

6. Given the concerns raised, there was a feeling that claims were worded too strongly, and that language should be toned down throughout.

We toned down the statement throughout the manuscript, and details are provided in the following reply to Reviewer 3’s comment of point 1.

Please note that, in light of the fact that the study's key claim currently seems insufficiently supported, we will send your revised manuscript out for re-evaluation and that eventual acceptance is not guaranteed.

Reviewer #1 (Recommendations for the authors):Our main comment is related to experimental design as the possibility that *Jeholornis* was not a specialist granivore or frugivore was not evaluated to the same extent as granivory and frugivory in your study. This third hypothesis would be important to address using multiple methods simultaneously. This is supported by the PCA data where there is overlap between the granivory and frugivory data points and the 'other diet' data points.The idea of a bird-plant link in the Early Cretaceous is interesting and could be spelled out more, along with further consideration of the uncertainties revealed in key studies e.g., Hulme 2002.Hulme, P.E., 2002. Seed-eaters: Seed Dispersal, Destruction and Demography. Seed dispersal and frugivory: Ecology, evolution, and conservation, p.257.However, this is secondary to our main comment above which is related to experimental design and the data incompletely supporting the conclusions proposed.We suggest two options to consider [### Note from the Senior Editor: our clear preference is Option 2 ###]:1) Narrow the scope of the conclusions to those which can be well supported by the data available i.e., a study showing that, for the seed components of the diet of *Jeholornis*, whether granivory or frugivory is most supported.2) Alternatively, we suggest to undertake a more holistic diet analysis based on multiple lines of evidence that tests hypothesis 3 and then using the results to inform the conclusions (which may or may not be related to granivory or frugivory).We hope you find the comments provided helpful.

Thank you very much for the comment and suggested solutions. We have replied to this in details in the previous “An Appraisal of Whether the Authors Achieved their Aims, and Whether the Results Support their Conclusions: ” part, since this is also the main point there. As we said there, there might be some misunderstanding about our study because that the word 'frugivorous' might be misleading to 'specialised frugivorous' to some readers including the reviewer. Actually Option 1 it seems what we claimed all along the manuscript that *Jeholornis* is at least partially frugivorous but not a specialist frugivore. We only need to rigorously rule out a granivorous explanation of the presence of seeds in the gut of *Jeholornis*, to demonstrate the partially frugivorous diet of *Jeholornis*, then we could indicate the occurrence of bird-plant interactions in Early Cretaceous. As we also replied in the previous part: the seed dispersal and frugivore ecology studies of the modern taxa show that, for most frugivores, fleshy fruits are a non-exclusive food resource, which is supplemented with other foods like animal prey and plants, and therefore avian frugivores occupy a wide range of diet space that is highly overlapping with some other diets. Therefore, this would not represent a narrowing of the scope of our conclusions ('at least partial frugivory') as concerned by the reviewer. However, to avoid this kind of understanding, as we stated in previous part: we use adjectives such as 'partial', 'seasonal' and 'opportunistic' as many as we could in the revised manuscript.

Knowing its holistic diet as the Option 2 may not be actually that necessary to our seed dispersal story, so that has almost no relation with our main goal in this study, but we agree that maybe some other readers will also be interested in a more detailed diet of this early avian lineage. Therefore, we conducted supplemental analyses by dividing 'other diets' further to test what diets *Jeholornis* possibly/impossibly had as supplements of frugivory, which would be testing the reviewer’s 'hypothesis 3'. We excluded some diets in the supplemental analyses and pointed out that what diet are possible to supplement fruits when the fruit resources are not available. More details were described in the previous part of reply. We expect these additional analyses would address the reviewer’s concern here.

Reviewer #2 (Recommendations for the authors):As far as I can tell, the descriptions and comparisons of skull morphology are detailed and well done. The argument that *Jeholornis* was not a seed cracker is strong, and the interpretation of *Jeholornis* as at least seasonally frugivorous is well-reasoned. It does nicely show that volant birds were recruited for endozoochory early. It will be a pleasure to see this published. Nonetheless, I do have some minor questions and concerns/recommendations about the botany and co-evolutionary implications that I think would make the manuscript stronger, along with some minor suggestions regarding the writing clarity that can be adopted at the authors' discretion.

Thank you very much for supporting our claim!

Line 60. Using plesiomorphic as an adverb strikes me as odd. It seems to me the meaning the sentence is unchanged if the word plesiomorphically is omitted and instead it reads "retaining an elongate bony tail"

Revised as suggested.

Line 64. In what way is *Jeholornis* the most representative jeholornithiform? Are other jeholornithiforms somehow less representative?

We have changed this to ‘…the most abundant jeholornithiform…’. Currently there are only two valid genera among the Jeholornithiformes: *Jeholornis* and *Kompsornis*. As we said in *Taxonomy of Jeholornis STM 3-8* in *Materials and methods* section, “The validity of another recently reported jeholornithiformes, *Kompsornis longicaudus* (Wang et al., 2020) needs more discussions since only one specimen is used to erect it, while no detailed comparisons have been done to the numerous specimens which have been assigned to *Jeholornis* before”.

Line 75. Consider substituting or adding "seed predation" alongside granivory. This widely used term emphasizes the ecological difference between feeding on seeds vs. other forms of herbivory as it kills a "whole plant."

Revised as suggested. We also added the term through the manuscript when necessary.

Line 77. Consider changing to "mutualistic co-evolutionary influence," or perhaps "co-diversification" is what you mean; however, see my comment about lines 342-343 below.

We revised it to be "mutualistic co-evolutionary influence".

Like 86. "Complete" or "precise" might be a better word choice than "accurate." Are previous reconstructions inaccurate? Perhaps they are just incomplete or poorly resolved, but not inaccurate.

We revised it to be “the most precise cranial reconstruction of a stem bird to date” as suggested.

Line 95. Because of its phylogenetic position… Because modifies a verb, in this case "studied" "Due to" means "caused by". Or, to put it another way, *Jeholornis* has been frequently studied because of its phylogenetic position.

We revised it to be “*Jeholornis* has been frequently studied and cited because of its key phylogenetic position” as suggested.

Line 130. I suggest "indicates that rostral fusion evolved earlier (or deeper in bird phylogeny) than previously thought." I also suggest that the authors point out what other characters it now precedes to emphasize the point that paleontology reveals the order of character changes along branches.

We revised it to be “Its occurrence in *Jeholornis* indicates that rostral fusion of premaxillae evolved phylogenetically deeper among birds than previously thought.” as suggested.

Line 225: I don't think the parentheses are necessary.

Parentheses were deleted as suggested.

Lines 260-263. I encourage the authors to describe the seeds, or at least to summarize the morphological diversity, number of types, (and perhaps their organization) if they were all previously described in O'Connor et al. (2018). What are their morphology, dimensions, and size variation? Do all of them have longitudinal striations? Based on the figures, these seeds appear to be quite large for Cretaceous angiosperms (Tiffney 1984; Eriksson et al. 2000; 2008). Is it more likely that these are seeds of a gymnospermous plant? I suspect so.

We added the summary of descriptions and more likely gymnosperm affinity of the ingested seeds in the paragraph discussing the identification of these seeds: “The alimentary contents preserved in *Jeholornis* were preliminarily described as ginkgo-like seeds (Zhou and Wu, 2006) and more likely to be gymnosperm due to their relatively large sizes, but have not been confidently identified with detailed comparisons with all the potential Early Cretaceous fruits/arils. In addition, although the poor preservation of these ingested seeds prevents any detailed taxonomic identification, three morphotypes have been grouped in previous studies based on size and shape: morphotype-1 in smaller size with a circular shape and curved striations, morphotype-2 in larger size with an oval shape, and morphotype-3 in similar size to morphotype-1 but with a strongly tapered pole (O’Connor et al., 2018).”

Lines 274-277. The hypothesis about seasonal frugivory in *Jeholornis* is both reasonable and interesting. Some indication of warm climate with seasonality of precipitation is supported elsewhere in the Jehol biota by Ding et al.'s (2016) analysis of fossil woods. It may be worth citing that work.

Thank you very much for the helpful citation information, it was added now.

Lines 305-315. Extant gymnosperms that use endozoochory for seed dispersal use a variety of accessory tissues, not just arils. Ginkgo, cycads, and some extinct groups have a fleshy sarcotesta (outer seed coat). An aril is a fleshy appendage of the seed coat or the funiculus. Some gymnosperms have fleshy subtending bracts (Ephedra), and some have fleshy accessory tissues such as bracts (Ephedra), cones (Juniperus), receptacle and epimatium (Podocarpaceae), or fleshy cones with numerous seeds (Juniperus). See Tiffney (1986, 2004); Herrera (1989); Lovisetto et al. (2012); Contreras et al. (2017); Nigris et al. (2021). I suggest Herrera (1989) as an earlier citation than Herendeen et al. (2017) for line 306.

Thank you very much for the helpful knowledge from the botanical view! This sentence was revised to be: “Although true fruits are only present in angiosperms, seed ferns and gymnosperms evolved functionally analogous fleshy-coated propagules (arils) and other fleshy accessory tissues much earlier (Tiffney, 1986, 2004; Herrera, 1989; Lovisetto et al., 2012; Contreras et al., 2017; Herendeen et al., 2017).” now.

Line 318. Why small? What does small mean? Small enough to fit in their mouth?

Yes we mean small enough for the early birds to swallow, in which way *Jeholornis* did it. We deleted the word ‘small’ to avoid confusion.

Line 321. I suggest "early and repeated" instead of "ancient". It appears to be clear that there are multiple origins of frugivory/omnivory within crown group birds, and it is not clear that the MRCA of crown-group birds was frugivorus (Felice et al. 2019). This lability in bird diet and repeated origin of frugivory might actually better support the authors' argument that diffuse coevolution between angiosperms and frugivorous birds contributed the KTR. There is now good evidence that volant birds have been repeatedly recruited by plants as endozoochorous seed dispeserers, even prior to the evolution of the MRCA of extant birds.

Thanks for the suggestion, we agree and revised this part to be “…the early and most likely repeated origin of frugivory in birds…” now.

Line 342-343. Can you provide a citation linking dispersal mode to diversification rate? Eriksson and Bremer (1992) in the journal 'Evolution' found no relationship between dispersal mode and diversification rate in angiosperms, although Carlo and Morales (2015) in the journal 'Ecology' showed evidence that frugivorous birds increase plant α diversity.

Thank you very much for the suggestion! We added Muller-Landau and Hardesty (2005), Dennis et al. (2007), Jordano (2014), Carlo et al. (2022), to support the mutualisms of frugivory and seed dispersal by animals here, which is one of the most studied mutualisms about biodiversity. We may also want to clarify here that our work supports the hypothesis that “bird-plant interactions played a role in the Cretaceous Terrestrial Revolution”, but we are not testing if it has a more important role than other factors e.g. insect pollination – we are far from reaching enough fossil information from either the bird or the insect side to test this at this moment. The mutualisms of frugivory and seed dispersal by animals among extant taxa are enough to confirm the benefit of the appearance of a new mutualistic mechanism like frugivory of birds in early KTR stage.

The study in Eriksson and Bremer (1992) about the extant dispersal mode to diversification rate are actually about comparisons of the influence of dispersal mode to other factors like insect pollination. In Eriksson and Bremer (1992), they were mostly claiming that (1) pollination systems and (2) life forms are more important factors than (3) animal dispersal. In addition, they emphasized that these three hypotheses are not mutually exclusive, and “several mechanisms in concert determine variation in diversification rate in plant”. They also emphasized that this evaluation was restricted for explaining present-day angiosperm diversification. We think the comparisons of influence of mostly insect involved biotic interactions and birds/mammals for present plants cannot be directly applied to the transitional, early KTR stage. Eriksson and Bremer (1992) also agreed that for the deeper evolutionary history, “a positive influence on diversification by animal dispersal, may have been important during Early Tertiary, but is nevertheless not detectable in a data set based on extant number of species.”.

We also added Carlo and Morales (2016) here as you mentioned (we think this is the paper you were talking about), since comparatively, the work in Carlo and Morales (2016) focusing on the speed and diversity of early successional forests resembled more to the early stage of KTR, while the angiosperms are also of much lesser abundance at that period.

Carlo TA, Morales JM (2016). Generalist birds promote tropical forest regeneration and increase plant diversity via rare‐biased seed dispersal. *Ecology* 97(7): 1819–1831.

Carlo TA, Cazetta E, Traveset A, Guimarães PR, McConkey KR. 2022. Fruits, animals and seed dispersal: timely advances on a key mutualism. *Oikos*, 2022: e09220.

Dennis AJ, editor. 2007. Seed dispersal: theory and its application in a changing world. CABI, Wallingford, UK.

Jordano P. 2014. Fruits and frugivory In: Gallagher RS, editor. Seeds: The Ecology of Regeneration of Plant Communities, 3rd edn. CABI, Wallingford, UK. pp. 18–61.

Muller-Landau HC, Hardesty BD. 2005. Seed dispersal of woody plants in tropical forests: concepts, examples, and future directions In: Burslem D, Pinard M, Hartley S, editors. Biotic interactions in the tropics: Their role in the maintenance of species diversity. Cambridge: Cambridge University Press. pp. 267–309.